# Silica Particles Trigger the Exopolysaccharide Production of Harsh Environment Isolates of Growth-Promoting Rhizobacteria and Increase Their Ability to Enhance Wheat Biomass in Drought-Stressed Soils

**DOI:** 10.3390/ijms22126201

**Published:** 2021-06-08

**Authors:** Anastasiia Fetsiukh, Julian Conrad, Jonas Bergquist, Salme Timmusk

**Affiliations:** 1Department of Forest Mycology and Plant Pathology, Swedish University of Agricultural Sciences (SLU), P.O. Box 7026, SE-75007 Uppsala, Sweden; stimmusk@hotmail.com; 2Swedish National Cryo-EM Facility, Science for Life Laboratory, P.O. Box 1031, SE-17121 Solna, Sweden; julian.conrad@scilifelab.se; 3Department of Chemistry, Uppsala University, P.O. Box 576, SE-75123 Uppsala, Sweden; jonas.bergquist@kemi.se; 4Bashan Institute of Science, 1730 Post Oak Ct, Auburn, AL 36830, USA

**Keywords:** plant drought stress tolerance, harsh habitat isolates, silica nanoparticles, exopolysaccharides, D-glucuronate (D-GA), hyperosmolarity, MALDI-TOF

## Abstract

In coming decades, drought is expected to expand globally owing to increased evaporation and reduced rainfall. Understanding, predicting, and controlling crop plants’ rhizosphere has the potential to manipulate its responses to environmental stress. Our plant growth-promoting rhizobacteria (PGPR) are isolated from a natural laboratory, ‘The Evolution Canyon’, Israel, (EC), from the wild progenitors of cereals, where they have been co-habituating with their hosts for long periods of time. The study revealed that commercial TM50 silica particles (SN) triggered the PGPR production of exopolysaccharides (EPS) containing D-glucuronate (D-GA). The increased EPS content increased the PGPR water-holding capacity (WHC) and osmotic pressure of the biofilm matrix, which led to enhanced plant biomass in drought-stressed growth environments. Light- and cryo-electron- microscopic studies showed that, in the presence of silica (SN) particles, bacterial morphology is changed, indicating that SNs are associated with significant reprogramming in bacteria. The findings encourage the development of large-scale methods for isolate formulation with natural silicas that ensure higher WHC and hyperosmolarity under field conditions. Osmotic pressure involvement of holobiont cohabitation is also discussed.

## 1. Introduction

Agriculture faces several challenges at the global level and, in coming decades, drought is expected to expand globally owing to increased evaporation and reduced rainfall, as well as or changes in the spatial and temporal distribution of rainfall [1]. The scientific community across the world is earnestly looking for novel solutions to enhance crop plant stress tolerance under limited resource availability, and several environmentally friendly solutions have shown huge potential but need to be optimised for wide-scale field application. One such solution includes strengthening plants’ natural defence systems with plant growth promoting rhizobacteria (PGPR). Predicting and controlling the rhizosphere has the potential to harness plant microbe interactions and restore plant ecosystem productivity, improve plant responses to environmental stress, and mitigate the effects of climate change [2,3]. The first report on PGPR-induced drought stress tolerance was published in 1999 [4]. The reported role in ERD15 and RAB18 gene expression has been repeatedly discussed and a robust set of *Arabidopsis*-associated microbiomes, varying in their ability to increase plant stress tolerance, has been identified [4,5,6,7,8,9,10]. While it is generally accepted that plant microbiome interaction under natural conditions is a complex and dynamic process, it is also known that the rapid changes in climatic pattern do not allow for adaptive and supportive crop plant microflora development. The plant microbiome, which is evolved with its host, can significantly contribute to its host’s stress adaptation [3,11,12]. Our wild barley (*Hordeum spontaneum*) rhizosphere isolate *Paenibacillus polymyxa A* 26 originates from a habitat exposed to various stress factors at the Evolution Canyon (EC) on the South Facing Slope (SFS) [13,14] (Appendix A). The *P*. *polymyxa* A26 Sfp-type 4-phosphopantetheinyl transferase deletion mutant strain (A26Sfp), which was enhanced in plant drought stress tolerance ability, was created [15]. The mutant, compared to its wildtype A26, is enhanced in biofilm exopolysaccharides (EPS) production and EPS D-glucuronate (D-GA) content [6,16,17]. This correlates with the strain with improved drought stress tolerance and enhanced biocontrol ability [6,15,16,17,18]. EC is a well described natural laboratory where microbes have co-habited with hosts over a long period of time [19]. Reciprocity between the organisms is generally accepted [20,21,22,23] and organisms that have co-evolved within the environment are more robust to environmental stress situations [24,25]. Microbes found in extreme habitats developed a different strategy to adapt to such conditions through evolution. Holobiotic complex relationships (interactions among many genes and environment) are becoming more and more accepted in developmental and evolutionary biology [20,21,22,23]. The ultimate goal of the PGPR application is to improve crop performance under natural conditions. This can be accomplished via the efficient formulation of technological understandings of a microbial framework’s adaptation to its environment [17,24,25,26]. The solid phase of any soil is composed of minerals (inorganic) and organic material. Minerals predominate in virtually all soils. The most abundant class of minerals are silicates, which have a substantial impact on soil characteristics, as their surfaces are inherently reactive, potentially forming strong or weak chemical bonds with soluble substances and further regulating the composition of the soil solution [27]. Silica nanoparticles (fraction of silicas <100 nm) have been studied as effective agents for phytoremediation and pathogen resistance, and as silicon fertilizer [27,28,29]. Our recent plant–nanointerface interaction research shows that metallic nanoparticles (TNs), along with native sand soil, promote A26 and A26Sfp effects on wheat plants [17]. Highly significant increases in plant biomass were observed under drought stress. Here, we tested the hypothesis that a sand soil fraction, silica nanoparticles (SNs), improves the PGPR effect. We used commercially well-characterised SNs [30,31] and showed that A26 and A26Sfp treatment with SNs increases PGPR wheat biomass enhancement abilities under drought stress. We further show that the particles trigger bacterial strain EPS D-glucuronate (D-GA) production, which increases the water holding capacity and osmotic pressure of the bacterial biofilm. These results are associated with bacterial cell elongation and the formation of bacterial cell clumps.

## 2. Results

Experimental setup (Table 1).

Experiments consisted of PGPR treatments applied in a randomized block design to plants of winter wheat (*Triticum aestivum* cv Stava) grown in plastic pots filled with 450 g of peat soil (Sol mull, Hasselfors). Peat soil contains traces of nutrients but no silica. Wheat seedlings were inoculated with the PGPR, as described under ‘Plant treatment’. The plant growth promoting effects of A26 and its mutant A26Sfp, as shown earlier, are linked to stress conditions, primarily to drought stress. Hence, to study the SN effect, the plants inoculated with PGPRs, with and without SNs, were exposed to drought stress, harvested, and dried. As peat soil composition interferes with the bacterial biofilm EPS D-GA content and mass spectrometric studies, the other sets of experiments were performed in hydroponic conditions and culture media. For the microscopic morphological assays, the bacteria were re-isolated from the plant rhizosphere and used, along with the PGPR laboratory cultures. Data were subjected to statistical analysis, as described under Data confirmation and validation.

### 2.1. Ludox^®^ TM50 Reassessment and Enhancement of Plant Biomass by PGPR with and without SN Treatment under Drought Stress

Commercial Ludox^®^ TM50 Sigma Aldrich particles’ [30,31] sizes and zeta potential were reassessed using Zetasizer (Malven Nano-ZS90, Malvern Instruments Ltd, Malvern, UK). Our results confirm the former evaluations and show that the SNs in the half-strength TSB medium possess a uniform shape (30 ± 3.1 nm), along with relatively favourable dispersibility (z potential −35 ± 0.64 mV). A total of 35% plant biomass improvement was observed by A26SfpSN inoculation compared to the A26Sfp inoculation, and there was a 30% improvement by A26SN compared to the A26 inoculation (*p* < 0.01) (Figure 1). Hence, the bacterial strain SN treatments resulted in significant plant shoot dry weight increases (Figure 1). The fate of the A26 and A26Sfp, with and without SNs, was followed using selection plates and PCR. In all, 10^3^–10^4^ bacterial cfu per gram of plant material after eight and 15 days of plant growth was detected (Table 2). When SN formulation was used, statistically similar colonisation numbers of 10^3^–10^4^ cfu at both time points were recorded (Table 2). Regression analysis indicates that there is a positive interaction between SN-formulated PGPR drought tolerance enhancement activity and EPS D-GA content (*r* > 0.82, *p* < 0.01).

### 2.2. EPS Production and EPS D-GA Content of the PGPR Grown with and without SNs

#### 2.2.1. Two Sets of Experiments Were Performed, and Both Were Culture Media Experiments

Strains A26, A26Sfp were grown in 1/2 TSB with 50 µg mL^−1^ silica particles (SN) at 30 ± 2 °C for 24 h. While the SNs did not have a significant impact on the bacterial number, the nanoparticles improved A26 and A26Sfp EPS production by 46% and 29%, respectively (Table 2). A26Sfp EPS production was 30–40% higher than its wildtype and A26Sfp SN treatment caused a further 20% increase in EPS production (Table 2).

The SNs improved the D-GA content of A26 and A26Sfp by 25% and 23%, respectively (Table 2). The D-GA content in A26Sfp was about 40% higher than in A26 (Table 2).

#### 2.2.2. Hydroponic Culture Experiments

To follow the A26, A26SN, A26Sfp, and A26SfpSN EPS and its D-GA content under more natural settings, the roots of the eight-day-old seedlings were carefully removed from the hydroponic solution. We aimed to collect the biofilms developed after the PGPR inoculation.

The biofilm bacteria were re-isolated by selective plating, confirmed by PCR as described earlier [14,32], quantified, and the supernatant polysaccharides were isolated and D-glucuronate content was recorded (Table 2). A26Sfp and its wildtype A26 (10^3^ per mL) were re-isolated in the root biofilms (Table 2).

Significantly higher EPS contents were detected in A26SN and A26SfpSN biofilms in comparison to A26 and A26Sfp biofilms (38 and 13%, respectively) (Table 2). While neither the A26 population density nor that of A26SN varied significantly, the D-GA content of A26SN and A26SfpSN was, respectively, about 40% and 35% higher (Table 2).

### 2.3. A26Sfp, A26SfpSN EPS MALDI Mass Spectrometry

We were interested to learn whether the relative increase in EPS titre, observed both in the bacterial cultures as well as in root biofilm assays (Table 2), was reflected in the polysaccharide mass spectrometric analysis. Hence, the A26, A26SN, A26Sfp, and A26SfpSN bacterial cultures were grown in half-strength TSB until the logarithmic growth phase, and polysaccharides were isolated. Typical patterns of the improved biofilm producers of A26Sfp and A26SfpSN abundance are shown in Figure 2. Comparative analysis of mass spectra indicates similar patterns of oligosaccharide chains (Figure 2). Quantification shows that, while A26SfpSN oligosaccharides with lower *m*/*z* values are decreased, the oligosaccharides with higher *m*/*z* values are increased up to 40%, reflecting the change in EPS titre (Table 3).

### 2.4. Silica Particles Induce Bacterial Elongation and Cell Aggregate Formation

Based on the results described above, it was hypothesised that the presence of SN changes the bacterial strain development and performance.

To study this, the bacteria were re-isolated from the plant rhizosphere after 15 days of growth. The identified A26, A26 Sfp, A26SN, and A26SfpSN bacteria were picked from the selected colonies, grown in half-strength TSB medium for 12 h, and visualised using a Celestron PentaView light microscope with an LCD screen. Additionally, A26 and A26Sfp cultures with and without SNs grown until the mid-logarithmic phase in culture media, as described above, were studied.

The PGPR cells’ morphology was studied using a hemocytometry chamber. A total of 40–50% of the A26SN and A26SfpSN rhizosphere isolates and A26SN and A26SfpSN media cultures were elongated. Without SNs, in A26 and A26Sfp, about 5% of elongated cells were formed (Figure 3). Around the elongated cells, bacterial clumps were formed (Figure 3 and Appendix A). Vigorous bacterial movements were recorded during the entire logarithmic phase of growth of A26SN and A26SfpSN bacteria, while the bacterial movement in the media without SNs was significantly slower after reaching mid-logarithmic growth (Appendix A).

We used Talos Arctica cryo-electron microscopy with a Falcon III camera in order to characterise the PGPR with and without SN treatment. Bacterial clumps were trapped on TED PELLA individual wells. No clumps and only individual cells were observed without SNs during the logarithmic phase of growth. Cryo-microscopic images of the washed and sonicated clumped cells show longer and larger cells with clear nucleoid structures (Figure 3).

### 2.5. EPS WHC and Osmotic Properties

A26 and its mutant A26Sfp, with and without SNs, were grown and EPS-extracted as described above. The addition of 0.6%, by volume, of the biopolymers significantly increased the WHC of peat soil. The SN treatment increased the water holding capacity (WHC) of A26 and A26Sfp strains by 14% and 16%, respectively (Table 4).

A26Sfp and its wildtype A26 osmotic properties, with and without SN treatment, were evaluated in half-strength TSB medium grown until mid-logarithmic phase. The contribution of the bacterial polymer to the osmotic pressure was calculated by subtracting the measured osmotic pressure of the pure medium. SN treatment increased the bacterial isolate polymer osmolarity about 15% (Table 4), and the effect was correlated with EPS production and plant biomass accumulation (*r* > 0.88, *p* < 0.01).

## 3. Discussion

Here, we evaluated the effect of commercial silica nanoparticles (SN) on EPS production by the harsh environment isolate A26 and its mutant A26Sfp. The silica nanoparticles exhibited good mono-dispersibility in the ½ TSB medium used in the study. It is well documented that particles are more likely to remain dispersed if the zeta potential is higher than 30 mV [33]. We showed that SNs further improved A26 and its mutant A26Sfp EPS production and enhanced their wheat drought tolerance ability. Although nanotechnology has revolutionized the world, there are safety measures to consider. The metallic nanoparticles (TNs) used by us earlier were prepared by Sol Gel technology [17]. Engineered nanoparticles are formed mainly by reduction of metal ions followed by functionalization of the NPs’ surface. Beside engineered NPs, such as metals, non-metals, metal oxides, and lipids, polymer NPs are formed naturally. Natural NPs are abundant in the atmosphere, hydrosphere, lithosphere, and even biosphere. They are formed by chemical, mechanical, thermal, and biological processes, weathering, and mechanical processes, combined with precipitation and colloid formation [34]. Our results show that nanoparticles have roles in the rhizosphere bacterial interactions with plants and may hold the key to the reproducible application of beneficial plant microorganisms in food security programs. NPs have unique physical and chemical properties and, due to their small size, are characterized by huge surface area [17,34]. Owing to their small size, NPs can potentially cause severe damage to ecosystems and human heath [35]. In the study, we applied amorphous silica particles with the understanding that SNs are always abundantly present in agricultural systems and will not interfere with the soils’ mineralogical, structural composition and biogeochemical cycles [17]. Silicas present a class of NPs with significantly higher potential for sustainable field applications than metallic NPs. While synthetic metallic NPs are irreplaceable in controlled environment applications, technologies using natural silicas should be encouraged.

Earlier, we showed that the stressful South Facing Slope (SFS) contains significantly higher populations of bacteria that contain 1-aminocyclopropane-1-carboxylate deaminase (ACCd), form biofilms, solubilize phosphorus, and are tolerant to osmotic stress [13] (Appendix A). It is evident from the number of reports that the PGPR strains used in the study promote plant fitness primarily under stress situations [14,15,17]. The growth-promoting experiments in the study reveal that SN treatments significantly enhance A26 and A26Sfp biofilm EPS production, thus improving the PGPR plant growth promoting ability (Table 1, Table 2 and Figure 1). It is well known that, even though bacteria are crucial for any biofilm formation, their quantification is insufficient for quantifying biofilms, as a biofilm matrix mainly consisting of EPS plays a crucial role. Owing to the huge complexity of EPS components, their detailed quantification in biofilms is a challenge. We have previously applied D-glucuronate (D-GA) as a proxy for biofilm comprehensive screening, as uronic acid is widely determined as representative of myco-polysaccharides in biofilms [6,17,36]. Here, we observed an obvious correlation of the PGPR EPS D-GA WHC, as well as its solution of osmotic pressure, with the corresponding strain’s ability to increase wheat drought stress tolerance (Table 4). Uronates, e.g., alginate and xantan, are charged sugars that form the basis of various polysaccharides with a high water absorption ability [37,38,39]. Uronic acid backbones lead to changes in other sugar backbones, which eventually result in alterations of their properties and bioactivity [40].

Unlike proteins, the biosynthesis of EPS is not template-driven, and polysaccharides occur as heterogeneous mixtures of high complexity. MALDI TOF has been developed as a successful tool for the analysis of biopolymers and is a valuable tool for comparison of polysaccharide profiles [41,42]. The ions produced are mostly single-charged molecular ions and the mass spectra can be used for distribution analysis of polysaccharides. Aqueous extracts can be successfully used without further purification [41,42]. The EPS production of the PGPR culture filtrates was compared to that of SN-induced strains. The results confirm the EPS titre results (Table 2). While the SN-induced A26 and A26Sfp low weight EPS chains show a relatively small increase, the higher weight EPS chain production increases by more than ten times (Figure 3, Table 3).

Here, we confirm the hypothesis that the treatment of harsh environment PGPR with silica nanoparticles, just as treatment with sandy soils, improves their ability to grow under drought conditions and enhances plant growth. This hypothesis was suggested by us owing to the growth promoting effect observed in sandy soils [17]. SN effects on plant and bacterial performance under various stress conditions were studied earlier, and one of the effects of bacterial count/ biomass increase is reported [27,28,29]. However, our results show that, while SNs improved the EPS production of both harsh environment strains used, the SN treatments did not increase bacterial count (Table 2). The results suggest the involvement of diverse mechanisms and indicate that isolates from harsh environments may benefit from their enhanced EPS production. Microorganisms in their native environments are subjected to various fluctuations in environmental conditions. It has been suggested that the ecological ‘success’ of EPS depends on their potential to beneficially influence their bacterial adaptation to the environment [43]. The EPS matrix serves as the microbial interface with the environment [44]. EPS molecules with high WHC [39,44] can mechanically protect against water stress and were suggested by us as a mechanism by which harsh environment rhizobacterial strains tolerate drought and enhance plant biomass [15,19]. Our results show that SN presence leads to EPS synthesis with higher D-GA content. The EPS protective layer with a high WHC on the root surface could re-establish water potential gradients under drought stress (Table 3). Even though it is a most probable mechanism earlier reported by several groups [6,34,45], it is likely not to be the only one, especially in view of the extreme habituation in the rhizosphere of wild progenitors of cereals. Production and accumulation of osmoregulatory compounds and the maintenance of low internal pressure via efflux pumps are additional potential mechanisms, all regulated by complex gene expression machinery. While it is known that the internal osmotic pressure in biofilms is primarily generated by the EPS, the possibility of a macromolecular osmolyte secreted and maintained by the cells, in response to the influence of osmotic pressure gradients on the growth characteristics of biofilms, remains largely underexplored. It is generally believed that reactions involved in gene translation are ionic-dependent and can be explained in electrostatic terms [46]. Studies indicate that there is a gap between the rough picture of the mechanism of ionic regulation and detailed behaviour of reactions at the molecular level [46,47,48,49,50]. While the complete signal transduction pathways of the cells’ response to osmotic challenge is far from being understood, the sequence of events starts with a signal from an osmo-sensory receptor. The signal is then transduced to transport systems, leading to the response of the cell and resulting in osmo-responsive changes in gene expression [15]. Increased EPS, which in turn increases internal hyperosmolarity, is likely to regulate the behaviour of EC PGPR cells. Microscope studies show that SNs induce changes in bacterial cells in laboratory cultures and the effect is persistent enough to observe it in bacterial cells re-isolated from the rhizosphere (Figure 3 and Appendix A). This indicates that SNs are associated with significant reprogramming in bacteria, and the treatment initiates novel physiological properties of the bacteria that are not present in cells without SN treatment.

One of the major concerns in PGPR field application is a lack of reproducibility [51]. Indeed, the PGPRs can excrete their effect via various mechanisms and their gene expression is dependent on constantly changing surrounding environmental factors [51]. Formulation agents which could be organic, inorganic, or liquid polymeric additives are used to increase the efficiency of PGPR [26]. It is crucial that the compounds used for formulation are recyclable, environmentally friendly, and support the beneficial effects of the inoculant to the crop plant. In the case of our PGPR originating from EC SFS, we will continue testing silicas for the most efficient formulation strategies. To grasp the basis of the complete system of A26 SN and A26SfpSN interaction, detailed functional genomic, proteomic, physiological, and physical studies in the framework of silica particles will be performed. A comprehensive analysis of regulatory genes and transcription factors based on the transcriptome of isolates exposed to osmotic stress treatments will be performed. Across this range of work, high resolution microscopy of the structure of the bacterial surface complex of biomolecules and cellular organelles will be applied to gain insight into how stress-tolerance is coupled with the morphology of the bacteria.

In summary, we show that the SN presence triggers the PGPR D-GA-containing EPS production. Due to increased EPS production, the bacterial biofilm matrix of WHC and osmotic pressure are increased, and cells morphology is changed. The PGPR originates from the silica-rich EC SFS, where it has been coevolved with wild barley rhizosphere. We speculate that the dynamics of the isolate are dependent on the EPS matrix microniches created around the cereal roots. This likely allows the PGPR to work as a functional unit with the wild progenitors of cereals under heat and drought stress. Whether or not silicas would function similarly for other PGPRs remains to be elucidated.

## 4. Material and Methods

### 4.1. Bacterial Growth and Culture Conditions

P. polymyxa A26 originates from wild barley rhizosphere from the South Facing Slope at the natural laboratory called the Evolution Canyon [13] (Table 1). The P. polymyxa A26 Sfp-type 4-phosphopantetheinyl transferase deletion mutant strain A26Sfp (A26∆sfp) was generated, as previously described [14,15,32,52]. Stock cultures were stored at −80 °C and streaked for purity on half-strength tryptic soy agar (1/2 TSA). All bacterial strains were grown until mid-logarithmic growth phase in half-strength tryptic soy broth (1/2 TSB; pH 6.2) at 30 ± 2 °C.

Cultures were centrifuged and pellets were re-suspended in PBS, as described earlier [32]. Finally, the cultures were adjusted to 106 cells/mL.

### 4.2. Bacterial Growth in the Presence of Nanoparticles

Commercial Ludox^®^ TM50 Sigma Aldrich particles were used in the study [30,31]. The hydrodynamic sizes and zeta potential of silica nanoparticles were examined by Zetasizer (Malven Nano-ZS90, UK). Suspensions of silica nanoparticles were dispersed by sonication prior to adding ½ TSB medium in order to minimize their aggregation. Strains A26 and A26Sfp were grown in 1/2 TSB with 50 µg mL^−1^ particles. The SNs were deagglomerated by ultrasound for 1–5 min and mixed with growth medium prior to bacterial inoculation. Culture density was determined by colony forming unit analysis (CFU). For the mock treatment, 1/2 TSB with 50 µg mL^−1^ SNs was used.

### 4.3. Plant Treatment

Winter wheat (Stava) seeds were surface sterilised by a 60 s wash in 99% ethanol, followed by a 6 min wash in 3% sodium hypochlorite solution, and a wash in 99% ethanol, and rinsed several times with sterilized water. For germination, the sterilised seeds were placed in Petri dishes with one sheet of filter paper moistened with 5 mL of distilled water. The dishes were kept in a dark incubator at 20 °C until germination. A germination paper (4.5 × 3.0 cm) was moistened with 10 mL of sterile water.

Uniformly germinated wheat seeds were grown in pots filled with SolMull Hasselfors soil (Sphagnum peat, Örebro, Sweden without silicas and additives, grain size fine-medium) and incubated in an MLR-351H (Phanasonic, Chicago, IL, USA) growth chamber at 24/16 °C (day/night) temperature, and 16 h photoperiod at 250 μmol m^−2^ s^−1^ (Table 1). On the fifth day of seedling growth, the PGPR inoculation was performed by adding 1 mL of bacteria to 10^6^ cells /mL (see bacterial growth and culture conditions). The soil moisture was adjusted to 75% water holding capacity. Soil moisture (12.5% of soil dry weight) was kept constant during the first 8 days of seedling growth. Soil volumetric water content was evaluated using 5TE soil moisture sensors (Decagon Devices, Inc., Pullman, WA, USA). The plants were exposed to seven-day-long drought stress after 8 days of growth. After eight and 15 days of growth, the A26 and A26Sfp bacteria were re-isolated from roots and shoots were dried at 105 °C to a constant mass, cooled, and weighed.

A26 and A26Sfp re-isolation was performed as described earlier [14,32]. Briefly, roots of ten randomly selected plants were carefully shaken and washed in sterile distilled water to remove loosely attached soil and to collect bacteria intimately linked to the plant root. Thereafter, the roots were homogenized and the content of endospore-forming bacteria was determined after heat treatment at 80 °C for 30 min. Selective TSA and PDA plates supplemented with antibiotic mixture were inoculated with 100 mL of these suspensions, corresponding to 10^−3^–10^−5^ g soil or plant rhizosphere material per plate. DNA was isolated from 1-day-old cultures on agar plates. Single colonies were resuspended to obtain a bacterial density of about 10^5^ colony forming units per mL suspension. A 0.3 mL aliquot of the bacterial culture was suspended in 4.7 mL of buffer (10 mM Tris-HCl, pH = 7.6, 50 mM KCl, 0.1% Tween 20). For lysing, the suspension was heated and immediately cooled on ice. The mixture was centrifuged at 6000× *g* for 5 min and the supernatant was used for PCR analysis. The bacterial DNA PCR analysis was performed with primers designed using Primer3 software targeting the 16S (16sA26 F and 16sA26 R identifying both A26 and A26Sfp,) and Sfp-type PPTase gene (Sfpdel F and Sfpdel R identifying only A26Sfp).

The other set of plants was grown hydroponically, as described earlier [53]. Briefly, uniformly germinated seeds were placed in plastic boxes (10 cm × 5 cm × 2.5 cm containing 1.5 L of nutrient solution, pH  =  5.8) and supported vertically. The hydroponic nutrient solution contained 0.5 mM CaCl_2_, 150 μM MgSO_4_, 1 mM KNO_3_, 0.5 mM NH_4_Cl, 2 μM Fe EDTA, 10 μM KH_2_PO_4_, 11 μM H_3_BO_3_, 2 μM MnCl_2_, 0.35 μM ZnSO_4_, 0.05 μM (NH_4_)_6_Mo_7_O_24_, and 0.2 μM CuCl_2_. The pH of the nutrient solution was adjusted to 6.0 and buffered with 1 mM MES. The nutrient solution was refreshed every third day. On the fifth day of seedling growth, the PGPR inoculation was performed by adding 1 mL of bacteria (10^6^ cells /mL), grown as described above (Table 1). After 8 days of growth, ten seedlings were used for root homogenization and bacterial identification and quantification, as described earlier [14]. EPS and D-GA content were evaluated from the seedlings grown in hydroponic culture (Table 1).

The peat soil and hydroponic experiments were repeated three times and were performed in four replicates, each consisting of 6 plants.

### 4.4. EPS and D-GA Content Evaluation

EPS were isolated from the A26, A26SN, A26Sfp, and A26SfpSN cultures and root biofilms. The bacterial cultures and root biofilms were prepared as described above. The root biofilm bacteria were re-isolated by selective plating, confirmed by PCR as described earlier [14,15], and quantified. EPS extraction was performed as described earlier [6,54]. Briefly, the biofilms were harvested by carefully rinsing of the roots with PBS. Bacterial cultures were diluted at a ratio of 1:5 with distilled water and centrifuged for 30 min at 17,600× *g* at 20 °C to separate cells. Then, EPS were extracted twice by precipitation, slowly pouring the supernatant into two volumes of isopropanol while stirring at 200 rpm. The filtered polysaccharide was suspended in a digestion solution consisting of 0.1 M MgCl_2_, 0.1 mg/mL DNase and 0.1 mg/mL RNase solution, and incubated for 4 h at 37 °C. Samples were extracted twice with phenol-chloroform, lyophilised using a freeze dryer (Virtis SP Scientific 2.0 USA), taken to the initial volume, and dialysed against distilled water. Analysis of uronic acid content was performed as described earlier by Mojica et al., 2007 [36], with small modifications. Briefly, the EPS pellets were weighed and dissolved in 200 µL of deionised water. Potassium sulfamic acid (4 M, pH 1.6) was added to the EPS solution and mixed by vortexing. Then, sodium tetraborate solution in concentrated sulfuric acid (0.0125 M) was added. The solutions were incubated for 5 min in a 100 °C water bath, cooled on ice for 3 min, and centrifuged at 2000 rpm for 10 min, after which 20 µL hydroxyophenol solution (0.15% *v/v*) was added to the supernatant. The solution was then mixed gently, and the absorbance was read at 520 nm. Each data point represents the average of twenty replicate measurements.

### 4.5. Light Microscopy

Randomly selected plants (see Plant treatment) were used to re-isolate the bacteria after 15 days of plant growth as described [14]. Following the identification, the single colonies were picked from the initial selective plates grown in half-strength TSB medium for 12 h and studied using a Celestron PentaView LCD Digital Microscope. Additionally, the PGPR strains were grown until mid-logarithmic phase of growth with and without SNs, as described above. A LCD Digital Microscope Celestron PentaView, CA USA was used to collect images and record videos of the bacterial growth. Cell elongation was evaluated using hemocytometer counting chambers.

### 4.6. Cryo-Electron Microscopy

A26, A26SN, A26Sfp, and A26SfpSN cultures were prepared as described above. At the mid-logarithmic phase of growth, the cultures were sieved in order to collect the bacterial clumps using PELCO prep-eze individual wells (Ted Pella Inc, Redding, CA, USA). The trapped bacterial clumps were then washed twice in PBS, sonicated, and adjusted to 10^3^ cells per mL. For comparison, from the cultures that did not form clumps, the bacterial pellets were washed twice in PBS and diluted to 10^3^ per mL.

A total of 3 µL of bacteria solution at a concentration of 10^3^ per mLwas applied to glow-discharged (40 s, 20 mA using Pelco easyGlow, Ted Pella Inc. Redding, CA, USA) C-Flat 200 mesh R2/2 (thick) grids. Using a Vitrobot Mark IV (Thermo Fisher Scientific, Waltham, MA, USA) operating at 4 °C and 100% humidity, grids were blotted for 4 s before being flash-frozen in liquid ethane.

All TEM data were collected with a Talos Arctica (Thermo Fisher Scientific) microscope operating at 200 kV and a Falcon III camera (Thermo Fisher Scientific) operating in integrating mode. Images were collected at 8700× nominal magnification using approximately 2 electrons/A2 total dose and a defocus of 30 µm.

### 4.7. A26Sfp, A26SfpSN EPS MALDI Mass Spectrometry

A26, A26SN, A26Sfp, and A26SfpSN bacterial cultures EPS were prepared as described above. MALDI analyses were performed with an upgraded Reflex II MALDI-TOF mass spectrometer (Bruker-Franzen Analytic Gmbh, Bremen, Germany). Angiotensin II (1045.5 u), ACTH clip 18–39 (2464.2 u), bovine insulin (5733.5 u), and equine cytochrome c (12,360.1 u) were used for external calibration of the mass spectrometer. All calibrants (1–5 pmol/µL) were dissolved in 0.1% trifluoroacetic acid (TFA) in ultrapure Milli-Q Plus water. 2,5-dihydroxy benzoic acid (DHB) was used as MALDI matrix. The matrix was dissolved in HPLC-grade acetonitrile (1 g/L for the seed layer) or in 0.1% TFA in acetonitrile/ultrapure water [1:1 *v/v*] (10 g/L, saturated for the sample/matrix mixture). All samples were prepared with the seed layer method. First, a matrix seed layer was created by depositing a droplet (1 µL) of a 1 g/L solution of matrix dissolved in acetonitrile on a highly polished, stainless steel sample probe. Thereafter, the 10 g/L matrix and sample solutions were mixed in a test tube 1:1 and a droplet (1 µL) of sample/matrix was deposited on the matrix seed layer. Samples were then left to dry totally in air. Samples were irradiated with a 337 nm nitrogen and a laserspot 10–20 µm in diameter. All spectra were acquired in the reflection mode at an accelerating voltage of 20 kV. Mass spectra were analysed using the software provided by Bruker Daltonics Life Science [55]. All spectra shown were calibrated using external calibration with a mass deviation of within 0.08%.

### 4.8. Osmolarity Assay

Strains A26, A26Sfp, A26SN, and A26SfpSN were grown until mid-logarithmic growth phase in half-strength tryptic soy broth, as described above, and the cultures were adjusted to 10^6^ cells/mL. The osmotic pressure of the solution was measured with an osmometer (Advanced Instruments freezing point osmometer, model 3250, advanced instruments Inc., Norwood, MA, USA). The contribution of the polymer to the osmotic pressure was calculated by subtracting the measured osmotic pressure of the pure medium.

### 4.9. Evaluation of A26 and A26Sfp EPS Water Holding Capacity (WHC)

Bacterial cultures were grown and harvested and polysaccharides were isolated, as described above. A total of 600 mg of A26 and A26Sfp EPS were mixed with 100 g peat soil (Sol mull, Hasselfors, Sweden) and determined at different wetting cycles for 24 h. The soil biopolymer mixture was then allowed to drain for 30 min and the weight after saturation was recorded. Following the wet weight estimation, the mixture was dried in an oven, cooled in a desiccator, and weighed again. Each experiment was carried out in triplicate. WHC was calculated as s percentage of water holding capacity = (gain in weight at saturation point/dry weight of the soil) × 100.

### 4.10. Data Confirmation and Validation

To ensure reproducibility, three biological replicates of every single PGPR treatment were performed, if not stated otherwise. Replicated data were studied for normal distribution and analysed by Unscrambler X15.1 and MiniTab17. One-way analysis of variance and a post hoc LSD test was used to identify treatments that were significantly different from controls. Linear regressions (Unscrambler X15.1) were used to determine the relationships between seedling biomass accumulation and D-glucuronic acid content.

## Figures and Tables

**Figure 1 ijms-22-06201-f001:**
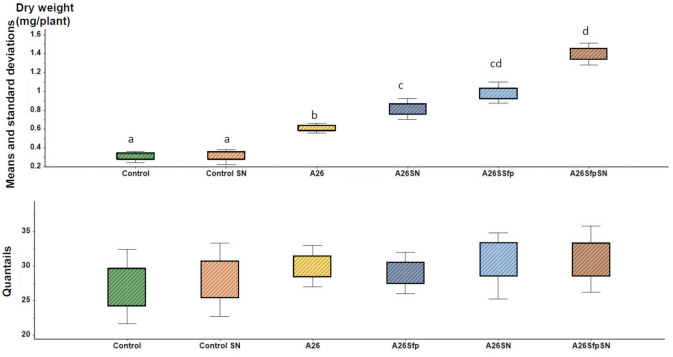
Seedlings weight analysis in soil assay. Plant dry weight means and standard deviations along box plot skewness of seedlings treated with A26, A26SN, A26Sfp, or A26SfpSN. ANOVA univariate analysis was performed and post hoc LSD tests were used to identify treatments significantly different from control (*p* < 0.05). Different letters indicate statistically significant differences.

**Figure 2 ijms-22-06201-f002:**
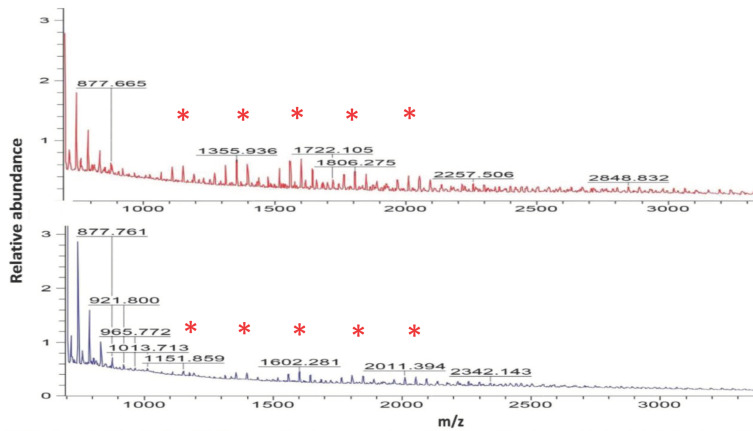
MALDI-TOF mass spectrometry of EPS extracts. Spectra of SN-induced A26Sfp (**A**) compared to A26SfpSN EPS extracts (**B**). Asterisks indicate oligosaccharide chain relative abundance compared in detail in Table (MALDI).

**Figure 3 ijms-22-06201-f003:**
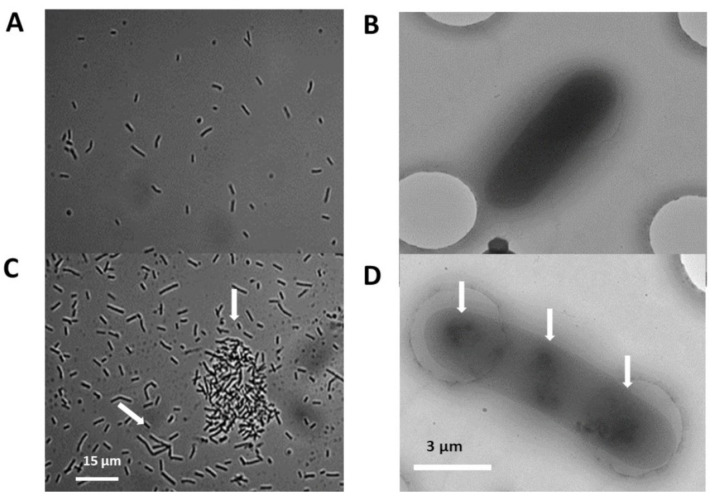
Micrographs of bacterial cells grown with and without silica nanoparticles (SN): Typical light- and cryo-electron microscopic images of A26 and A26Sfp cells (**A**,**B**), and A26SN and A26SfpSN cells (**C**,**D**). Arrows indicate cell elongation, bacterial aggregate formation, and changes in nucleoid structure. See Section 4.

**Table 1 ijms-22-06201-t001:** Summary of experimental assays.

Time (Days)	Peat Soil	Hydroponic System	Bacterial Cell Culture
1–5	Seedlings growth		
5	Inoculation with A26, A26SN, A26Sfp and A26SfpSN		EPS, D-GA assay
8	Rhizosphere population assay	Rhizosphere population assay/ EPS, D-GA assay	
8–15	Drought stress treatment		
15	Harvest/plant biomass analysis		

**Table 2 ijms-22-06201-t002:** EPS and D-GA assessment in relation to bacterial growth.

	Bacterial Population Log CFU/mL ^1^	EPS (µg/mL)	D-GA (10^−3^ µg/mL)
1/2 TSB cultures			
A26	9.00 ± 0.4 ^a^	11 ± 2 ^a^	0.3 ± 0.03 ^a^
A26SN	8.69 ± 0.37 ^a^	14 ± 2.1 ^b^	0.4 ± 0.04 ^b^
A26Sfp	9.0 ± 0.4 ^a^	14.5 ± 2.3 ^b^	0.7 ± 0.07 ^c^
A26SfpSN	8.89 ± 0.4 ^a^	18 ± 2.4 ^c^	0.9 ± 0.08 ^d^
Hydroponic culture			
A26	3.15 ± 0.15 ^b^	10 ± 1.5 ^a^	0.3 ± 0.07 ^a^
A26SN	3.09 ± 0.17 ^b^	14.8 ± 1.1 ^b^	0.5 ± 0.08 ^b^
A26Sfp	3.08 ± 0.13 ^b^	14.6 ± 1.3 ^b^	0.8 ± 0.8 ^c^
A26SfpSN	3.02 ± 0.15 ^b^	18 ± 1.9 ^c^	1.17 ± 0.2 ^d^
Root wash Control	1.19 ± 0.11 ^a^	0.08 ± 0.16 ^d^	0.05 ± 0.01 ^e^
Peat soil			
A26	3.02 ± 0.15 ^b^	ND	ND
A26SN	3.07 ± 0.13 ^b^	ND	ND
A26Sfp	3.09 ± 0.17 ^b^	ND	ND
A26SfpSN	3.11 ± 0.13 ^b^	ND	ND

^1^ The bacteria were re-isolated and confirmed by PCR [14,32]. ^a,b,c,d,e^ Different letters indicate statistically significant differences (*p* < 0.05). See Section 4.

**Table 3 ijms-22-06201-t003:** *Paenibacillus polymyxa* A26Sfp and A26SfpSN culture filtrate polysaccharide MALDI-TOF analysis.

№	*m*/*z*	Intensity		Percentage Increase
		A26	A26SN	
1.	789	141,907.458	90,814.536	55
2.	833	88,808.125	63,286.470	40
3.	877	72,049.000	63,445.000	14
4.	1109	45,547.000	56,170.000	19
5.	1151	46,583.000	51,814.284	10
6.	1194	43,345.000	44,568.000	3
7.	1272	36,336.000	46,047.000	21
8.	1356	44,012.000	60,857.115	28
9.	1398	41,128.571	53,392.158	23
10.	1440	35,220.000	38,714.000	9
11.	1476	30,765.000	40,044.000	23
12.	1518	34,532.000	48,457.729	29
13.	1560	39,376.080	59,106.113	39
14.	1602	41,044.232	59,972.852	32
15.	1644	37,606.941	48,760.800	23
16.	1806	32,269.259	38,433.907	16
17.	1848	30,604.477	33,294.678	8
18.	2052	27,555.650	32,217.000	14
19.	2173	25,381.000	24,539.000	5
20.	2215	26,874.000	28,241.000	5

See Section 4.

**Table 4 ijms-22-06201-t004:** Effect of bacterially produced biopolymers on osmotic properties and WHC. Different letters indicate statistically significant differences (*p* < 0.05).

	Fold Change	Strain	Osmolarity mOsm/kg	WHC%	EPS D-GA ** (E-03 µg/mL)
Plant Dry Weight Improvement under Drought Stress *	2	A26	285 ± 13 ^a^	42 ± 3 ^a^	0.3 ± 0.07 ^a^
	2.7	A26SN	299 ± 10 ^b^	48 ± 3 ^b^	0.5 ± 0.08 ^b^
	3.3	A26Sfp	325 ± 15 ^c^	59 ± 3 ^c^	0.8 ± 0.07 ^c^
	4.6	A26SfpSN	347 ± 13 ^d^	71 ± 3 ^d^	1.17 ± 0.2 ^d^

* See Figure 1 and Section 4. ** data for hydroponic culture are presented.

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
