# Peer review of "Silica Particles Trigger the Exopolysaccharide Production of Harsh Environment Isolates of Growth-Promoting Rhizobacteria and Increase Their Ability to Enhance Wheat Biomass in Drought-Stressed Soils"

_ijms, 2021, doi:10.3390/ijms22126201_

Round 1
Reviewer 1 Report
The manuscript by Fetsiukh and co provides an interesting, and for me personally, a new perspective on the role of silica particles in microbe-plant interactions.
However, I think the authors could make some improvement to their manuscript which would greatly help the reader.
Firstly, some of the sentences in the introduction (e.g., the second sentence) is quite long. I had to read it several times to get the sense of it. Secondly, many of the sentences are unclear. What, for example, does the following mean: 'It is known that the rapid nature of changes in climate pattern does not allow the adaptive and supportive crop plant microflora development but rather the plant microbiome, which is evolved with its host and can significantly contribute to its host's stress adaptation.' I'm just not sure what the authors are getting at. For someone who is not so familiar with this area of study, it makes for a hard read. I recommend a re-write of this and multiple other sentences throughout the manuscript to de-specialize the text.
I would also like to see more background information to the mutant Sfp in the manuscript, without having to run to the literature. Specifically, how does the mutation enhance polysaccharide production? Is this known?
The start of the results section is pretty hard to read. Specifically, the approach and the expectations could be laid out before the results are stated. Otherwise, one is left confused until the discussion, whereupon one needs to re-read the results section again.
What is WHC excretion? (Line 167) Should not this be about the excretion of molecules important for WHC? (A capacity cannot be excreted.)
Another aspect which contributes to confusion is the extensive use of acronyms. Some of these are not explained, such as ACCd and SFS (Line 200). Some seem interchangeable, such as SNs and NPs, or? Others may not be necessary. I spent a lot of time running around the manuscript searching for acronym explanations.
In Figure 2, if nanoparticles do not affect growth, why do many more cells appear in C as in A? That gives a misleading impression.
Line 128 Is the reference to Figure 3 supposed to be Table 3? As for Table 3, there are way too many decimal places in the data .. is the mass spec really that accurate? Also, it looks like one of the periods is replaced with a comma (No. 10, A26).
Line 238. The authors claim that the data suggests the involvement of diverse mechanisms. But I don't necessarily see it that way. It could also be some simple mechanism, such as SN increasing cell access to oxygen, or absoption of some nutrient (e.g., phosphate). Both the limitation of phosphate and excess of oxygen can activate EPS production in a variety of rhizobia. Is peat soil high in P and low in oxygen?
Is the importance of natural silica in agricultural production recognized in the literature, or is this a specialty of EC isolated PGPRs? The authors point to this question in their last sentence in the discussion, but if there is something known about this from the literature, I would like to see it included in this manuscript.
Author Response
We are grateful to the reviewer for the kind interest in our work and constructive comments. We hope we have managed to successfully correct the manuscript.
Firstly, some of the sentences in the introduction (e.g., the second sentence) is quite long. I had to read it several times to get the sense of it. Secondly, many of the sentences are unclear. What, for example, does the following mean: 'It is known that the rapid nature of changes in climate pattern does not allow the adaptive and supportive crop plant microflora development but rather the plant microbiome, which is evolved with its host and can significantly contribute to its host's stress adaptation.' I'm just not sure what the authors are getting at. For someone who is not so familiar with this area of study, it makes for a hard read. I recommend a re-write of this and multiple other sentences throughout the manuscript to de-specialize the text.
Response: several sentences in the Introduction and Discussion are rewritten for clarity and to be more informative (highlighted in yellow).
I would also like to see more background information to the mutant Sfp in the manuscript, without having to run to the literature. Specifically, how does the mutation enhance polysaccharide production? Is this known?
Response: corrected in the manuscript line 55-57. We however could not include more background information, which would risk the focus of the manuscript.
The start of the results section is pretty hard to read. Specifically, the approach and the expectations could be laid out before the results are stated. Otherwise, one is left confused until the discussion, whereupon one needs to re-read the results section again.
Response. The chapter Experimental setup should be included as an introduction to the Results chapter, making it clear and understandable from the beginning.
What is WHC excretion? (Line 167) Should not this be about the excretion of molecules important for WHC? (A capacity cannot be excreted.)
Response: corrected in the manuscript
Another aspect which contributes to confusion is the extensive use of acronyms. Some of these are not explained, such as ACCd and SFS (Line 200). Some seem interchangeable, such as SNs and NPs, or? Others may not be necessary. I spent a lot of time running around the manuscript searching for acronym explanations.
Response: corrected in the manuscript, the abbreviation SFS was explained in the Introduction. However, it is beneficial that we include both explanations as part of the Discussion.
In Figure 2, if nanoparticles do not affect growth, why do many more cells appear in C as in A? That gives a misleading impression.
Response: One of the features of SNs is that they cause bacterial cells to grow clumpy. While the untreated cells are rather homogeneously distributed on the microscopic field, the occasional clumps appear on the SN treated sample field .
Line 128 Is the reference to Figure 3 supposed to be Table 3? As for Table 3, there are way too many decimal places in the data .. is the mass spec really that accurate? Also, it looks like one of the periods is replaced with a comma (No. 10, A26).
Response: A maximum of three to four decimal places are requested for mass -spec results.
Line 238. The authors claim that the data suggests the involvement of diverse mechanisms. But I don't necessarily see it that way. It could also be some simple mechanism, such as SN increasing cell access to oxygen, or absoption of some nutrient (e.g., phosphate). Both the limitation of phosphate and excess of oxygen can activate EPS production in a variety of rhizobia. Is peat soil high in P and low in oxygen?
Response: The sentence refers to the generally observed mechanism of SNs i.e. increased number of cells. Our results do not confirm this, and we would suggest that other mechanisms are involved.
Is the importance of natural silica in agricultural production recognized in the literature, or is this a specialty of EC isolated PGPRs? The authors point to this question in their last sentence in the discussion, but if there is something known about this from the literature, I would like to see it included in this manuscript.
Response: It is generally known that soils are full of silicas and that silica nanoparticles are beneficial in various ways. Natural nanosilica- based stimulation may trigger beneficial effects of PGPRs and can improve crop yield. However, with sand soil as a growth substrate, the problem with nutritional deprivation will occur, as sands are poor in nutritional elements. Likewise, it is hard to predict the efficiency of heterogeneous particles of varying nature, size and amount in sand soils. Still, the presence of naturally existing nanoparticles must be considered when applying PGPRs.
Reviewer 2 Report
General comments: the manuscript is interesting and well structured. However, it needs to be thoroughly improved in methods. Particle characterization is not included. Furthermore more few minor points need to be addressed.
Introduction
Line 71. “We used commercial well characterised SNs [27, 28]”. This is definitely not enough for particle characterization. In the last 10 years several steps forward have been made both in methods for physico-chemical characterization and, consequently, either the information requested for publication need to be more accurate.
Line 72. Please fix text format/size.
Discussion
Lines 252-259. Please fix text format/size.
Methods
Line 322. As previously observed, this is not enough. Particle size, shape, z-potential, hydrodynamic range in water and in the media used for the experiments are fundamental parameters that have to be introduced and commented in the manuscript. Commercial characterization or references from 2011 cannot be representative of the particle batches utilized in current experiments.
Line 350. Please correct the typo.
Line 352. Please correct the typo.
Figures and tables
Please check the statistics in Figure 1. Not sure about “cd” and “e”
Figure 3. “Asterisks indicate oligosaccharide chain relative abundance compared in detail in Table (MALDI)”. Where are the asterisks?
Table 1 need to be improved. This format is totally illegible.
Author Response
We are grateful to the reviewer for the constructive comments and for drawing our attention to the crucial issues. We hope that we have managed to resolve the issues in a way that is sufficient.
Line 72. Please fix text format/size.
Response: Corrected in the manuscript
Lines 252-259. Please fix text format/size.
Response: Corrected in the manuscript
Line 322. As previously observed, this is not enough. Particle size, shape, z-potential, hydrodynamic range in water and in the media used for the experiments are fundamental parameters that have to be introduced and commented in the manuscript. Commercial characterization or references from 2011 cannot be representative of the particle batches utilized in current experiments.
Response: The particle size and Z potential has been re-examined. The results and definitive discussion are included in the following chapters (see below).
Line 100-103
Line 300-306
Line 176-181
Line 350. Please correct the typo.
Response: Corrected in the manuscript
Line 352. Please correct the typo.
Response: Corrected in the manuscript
Please check the statistics in Figure 1. Not sure about “cd” and “e”
Response: Thank you for the comment. It is corrected.
Figure 3. “Asterisks indicate oligosaccharide chain relative abundance compared in detail in Table (MALDI)”. Where are the asterisks?
Response: Unfortunately, the pdf version deleted the asterisks, which is an important part of the paper. The new Figure has been uploaded.
Table 1 need to be improved. This format is totally illegible.
Response: Table 1 has been Improved, focusing on significance and avoiding redundancy.
Round 2
Reviewer 2 Report
The manuscript has been improved. There is still a minor concern to fix in Figure 3.
Figure 3. asterisks are still missing. I suggest to introduce the asterisks directly in the Figure, in order to maintain them in the final version.